# A bed nucleus of stria terminalis microcircuit regulating inflammation-associated modulation of feeding

Yong Wang [1,2], JungMin Kim [1], Matthew B. Schmit[1,3], Tiffany S. Cho[1], Caohui Fang[1] & Haijiang Cai [1,4]

Loss of appetite or anorexia associated with inflammation impairs quality of life and increases morbidity in many diseases. However, the exact neural mechanism that mediates inflammation-associated anorexia is still poorly understood. Here we identified a population of neurons, marked by the expression of protein kinase C-delta, in the oval region of the bed nucleus of the stria terminalis (BNST), which are activated by various inflammatory signals. Silencing of these neurons attenuates the anorexia caused by these inflammatory signals. Our results demonstrate that these neurons mediate bidirectional control of general feeding behaviors. These neurons inhibit the lateral hypothalamus-projecting neurons in the ventrolateral part of BNST to regulate feeding, receive inputs from the canonical feeding regions of arcuate nucleus and parabrachial nucleus. Our data therefore define a BNST microcircuit that might coordinate canonical feeding centers to regulate food intake, which could offer therapeutic targets for feeding-related diseases such as anorexia and obesity.

---

[1] Department of Neuroscience, University of Arizona, Tucson, AZ 85721, USA. [2] Department of Physiology and Pathophysiology, Xi'an Jiaotong University Health Science Center, Key Laboratory of Environment and Genes Related to Diseases, Ministry of Education, Xi'an 710061, P.R. China. [3] Graduate Interdisciplinary Program in Neuroscience, University of Arizona, Tucson, AZ 85721, USA. [4] Bio5 Institute and Department of Neurology, University of Arizona, Tucson, AZ 85721, USA. Correspondence and requests for materials should be addressed to H.C. (email: haijiangcai@email.arizona.edu)

Inflammation-associated loss of appetite or anorexia during acute diseases may help maintain body integrity and modify immunocompetence for pathogen elimination, but inflammation-associated anorexia during chronic diseases such as cancer and HIV-infection has a negative impact on recovery and treatment success, decreases the quality of life and increases morbidity[1–3]. This type of anorexia has been well modeled in animal studies in which feeding suppression was induced by peripheral administration of pro-inflammatory cytokines such as interleukin-1β (IL-1β) and stimulants of the release of cytokines such as lipopolysaccharide (LPS) (for example, refs. [4–8]). Prior studies suggested that the central nervous system is the major target and mediator of the pro-inflammatory cytokines' anorexic effect during the infection or tissue damage that accompany diseases[3,6,8]. However, the exact brain regions and the underlying neural mechanisms that regulate inflammation-associated anorexia remain to be determined.

Peripheral administration of IL-1β, LPS, or other pro-inflammatory cytokines induces c-Fos expression, a cellular marker widely used to indicate neural excitation, in various brain regions especially the arcuate nucleus (ARC), lateral parabrachial nucleus (LPB), central nucleus of amygdala (CEA), and bed nucleus of stria terminalis (BNST)[9–13], many of which have been classically associated with the regulation of food intake. However, except neurons in LPB which have been suggested to play a mild role in rescuing anorexia induced by LPS[14], neurons in ARC and CEA tested so far have no effect in regulating inflammation-associated anorexia. For example, activation of the ARC Agouti-related protein (AGRP) neurons potently promote feeding but cannot restore LPS-induced anorexia[15]. Instead, LPS completely suppresses AGRP neuron-induced food intake[16]. Silencing a specific subset of CEA neurons blocks anorexia induced by diverse anorexigenic signals but cannot rescue the anorexia induced by LPS[17].

Compared to the neurons in ARC, LPB, and CEA, the role of BNST neurons in feeding is relatively unexplored. Several recent studies suggested that BNST neurons might play a role in regulating feeding. Activation of the inhibitory γ-aminobutyric acid-releasing (GABAergic) nerve terminals in BNST projected from ARC AGRP neurons or somatostatin neurons in the tuberal nucleus (TN) increases food intake[18–20]. In contrast, optogenetic activation of the pathway from the BNST neurons labeled by vesicular GABA transporter (VGAT) to the lateral hypothalamus (LH), a crucial neural substrate for feeding, also induces food intake[21]. Additionally, pharmacological studies which infused drugs in BNST, as well as anatomical studies also suggested that neurons in BNST subregions might play specific roles in feeding regulation during stress or other conditions[22–24]. However, which type of BNST neurons and whether neurons in BNST really regulate feeding or play a role in inflammation-associated anorexia is unknown.

To dissect the BNST neural circuits and identify the BNST neurons that regulate feeding, we searched the Allen Brain Atlas (http://mouse.brain-map.org)[25] for genetic markers enriched in subnuclei of BNST. We found that protein kinase C-delta (PKC-δ) is a marker that labels neurons exclusively in the oval region of BNST (ovBNST), which is a region previously demonstrated to include neurons activated by LPS or IL-1β[9,10]. Here we show that ovBNST PKC-δ neurons are activated by peripheral administration of IL-1β and LPS, and chemogenetic silencing of these neurons can effectively attenuate the inflammation-associated anorexia. Importantly, our results demonstrate that, besides inflammation-associated anorexia, these neurons mediate bidirectional control of general feeding. Our experiments also reveal that these neurons inhibit LH-projecting neurons in ventrolateral BNST, which antagonize the anorexic effect of activating ovBNST

PKC-δ neurons. Moreover, these neurons receive inputs from the canonical ARC, LPB, and CEA brain regions of feeding. Thus, our study identifies a unique BNST microcircuit that might function as a central hub in integrating the distributed feeding circuits into a hierarchy of brain structure for feeding regulation.

## Results

**ovBNST PKC-δ neurons regulate anorexia of IL-1β or LPS.** Firstly, we tested if inflammatory signals could activate specific type of neurons in BNST by double immunostaining for Fos and neuronal markers after intraperitoneal (IP) injection of IL-1β or LPS. Both IL-1β and LPS induced Fos expression preferentially in the ovBNST PKC-δ positive neurons (Fig. 1a–d and Supplementary Fig. 1). Notably, the majority (~80%) of the Fos+ neurons activated by IL-1β or LPS in ovBNST are positive for PKC-δ staining (Fig. 1d). We also found that the ovBNST PKC-δ neurons are preferentially activated by IP injection of another inflammatory cytokine tumor necrosis factor alpha (TNFα), but not the satiety peptide cholecystokinin (CCK) (Supplementary Fig. 2). We next investigated whether the activation of ovBNST PKC-δ neurons is required for inflammation-associated anorexia induced by IL-1β or LPS. To do this, we stereotaxically injected Cre-recombinase dependent adeno-associated virus (AAV) encoding inhibitory (hM4Di) designer receptors exclusively activated by designer drugs (DREADDs)[26] into the ovBNST of PKC-δ-Cre mice[27] bilaterally. After ~4 weeks to allow for virus expression and mice recovery, we injected the hM4Di ligand clozapine-N-oxide to silence the ovBNST PKC-δ neurons. 40 min after CNO or drug injection, we then measured the amount of food intake in a 20-min feeding session in mice fasted for 24 h (Fig. 1e). Sex and IP injection of CNO in animals expressing mCherry did not affect the amount of food intake in mice when normalized to their body weight (Supplementary Fig. 3). Brain slice electrophysiology recordings validated that firing of the ovBNST PKC-δ neurons expressing hM4Di can be silenced by CNO (Fig. 1f). We found that chemogenetic silencing of the ovBNST PKC-δ neurons significantly attenuates the anorexia induced by IL-1β or LPS (Fig. 1g). These results demonstrate that ovBNST PKC-δ neurons play an important role in mediating inflammation-associated anorexia.

Particular physiological properties of BNST neurons give rise to distinctive discharge patterns and might play important roles during behavior[28–30]. We therefore examined the electrophysiological properties of the ovBNST PKC-δ neurons in brain slices. BNST neurons have been classified into four different types based on their firing pattern in response to current injections (Supplementary Fig. 1J)[28,29]. While all four types of cells are observed, type III cells with a characteristic delay of firing in response to suprathreshold depolarizations are most abundant in ovBNST PKC-δ population (Supplementary Fig. 1K).

**Activation of ovBNST PKC-δ neurons suppresses feeding.** We next asked the question whether the ovBNST PKC-δ neurons regulate general feeding behaviors. To do this, we stereotaxically injected Cre-dependent AAV encoding ChR2[31] into the ovBNST of PKC-δ-Cre mice and implanted ferrule fibers above the ovBNST bilaterally. Photo-stimulation with brief light pulses reliably induced action potentials in ovBNST PKC-δ neurons at multiple frequencies (Fig. 2a, b). After a recovery and virus expression period of ~4 weeks, we coupled the ferrule fibers with a blue laser (473 nm) and optogenetically activated the ovBNST PKC-δ neurons of fed mice. We found that optogenetic activation of the ovBNST PKC-δ neurons strongly suppressed the total amount of food intake (Fig. 2c and Supplementary Fig. 4a). The strong feeding inhibition was also observed in mice fasted for 24 h

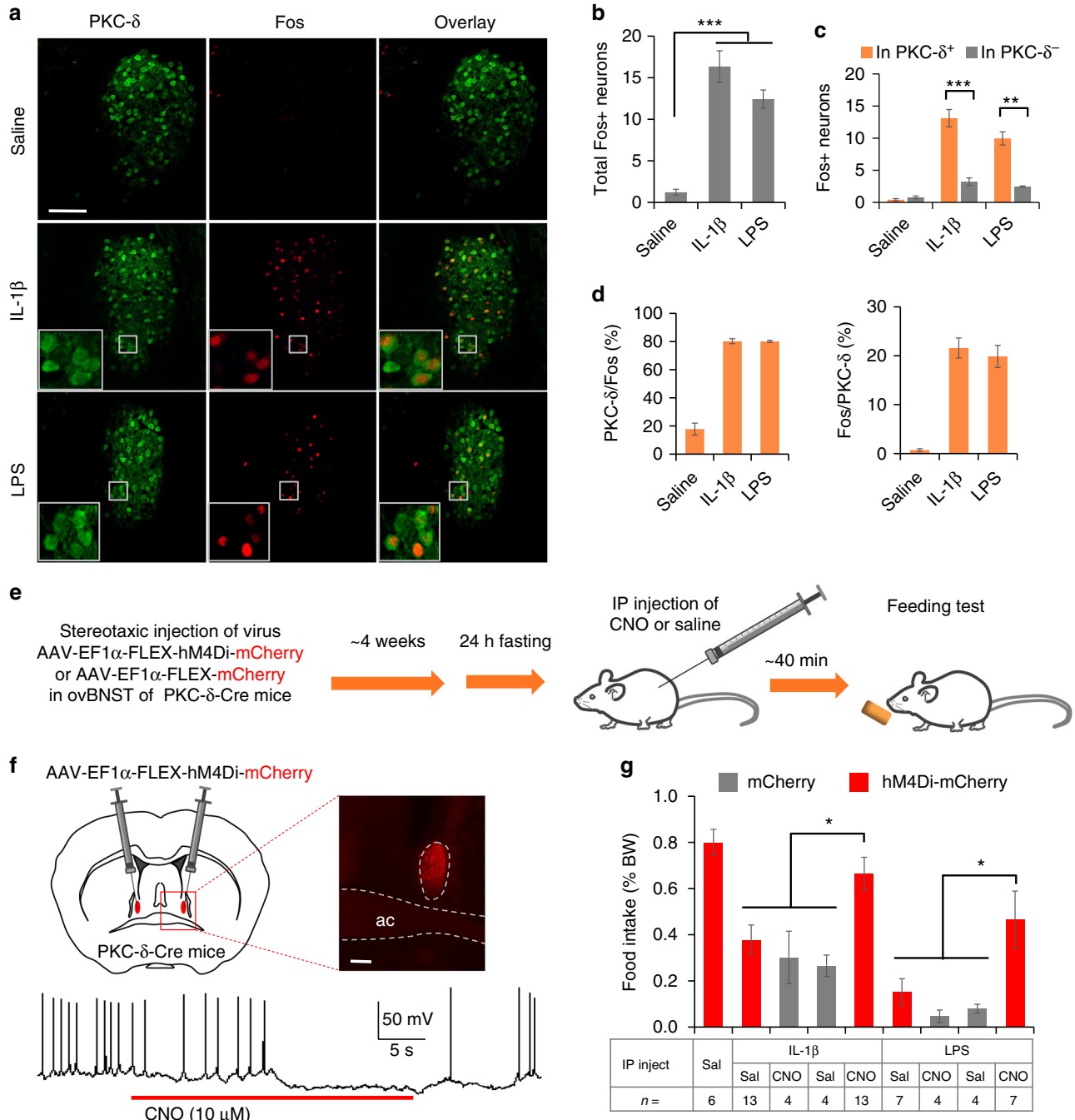

**Fig. 1** ovBNST PKC-δ neurons regulate the anorexia induced by IL-1β or LPS. **a–d** Representative histology (**a**) and quantification (**b–d**) of Fos-like immunoreactivity in each ovBNST brain section after IP injection of saline, IL-1β, or LPS. One-way ANOVA with post-hoc Bonferroni $t$-test ($F_{2,18} = 59.2$, $p < 0.001$, **b**), unpaired $t$-test (IL-1β, $t_{10} = 6.74$, $p < 0.001$; LPS, $t_4 = 7.28$, $p = 0.002$, **c**), $n = 10$ animals injected with saline, 6 animals injected with IL-1β, 3 animals injected with LPS. **e** Experiment procedure for feeding test after chemogenetic silencing of ovBNST PKC-δ neurons. **f** Expression of hM4Di in ovBNST PKC-δ neurons was achieved by stereotaxic injection of AAV-EF1α-FLEX-hM4Di-mCherry in ovBNST (up). Brain slice electrophysiological recording showed that the firing of ovBNST PKC-δ neurons expressing hM4Di-mCherry can be silenced by CNO (10 μM) (bottom). ac anterior commissure. **g** Food intake normalized to the body weight (% BW) in 24-h fasted animals after IP injection of different agents. Two-way ANOVA with post-hoc Bonferroni $t$-test showed a significant effect after CNO silencing of the ovBNST PKC-δ neurons. $F_{(1,30)} = 6.34$ (IL-1β), $F_{(1,18)} = 7.11$ (LPS), $n = 4–13$ animals (indicated below each group). Data are mean ± s.e.m. Scale bars, 100 μm. *$p < 0.05$, **$p < 0.01$, ***$p < 0.001$. Source data are provided as a separate file

(Fig. 2d and Supplementary Fig. 4b, d). The level of feeding suppression was dependent on the frequency of the stimulation (Fig. 2d). Light activation of the ovBNST PKC-δ neurons increased the latency to approach the food (defined as the time at which the animal's nose touches the food and starts eating) and

decreased the number of feeding bouts and the total amount of time spent on feeding (Fig. 2e–g). Photo-stimulation of ovBNST PKC-δ neurons strongly suppressed food intake in both male and female animals, and we did not observe any significant difference between male and female animals after virus expression or light

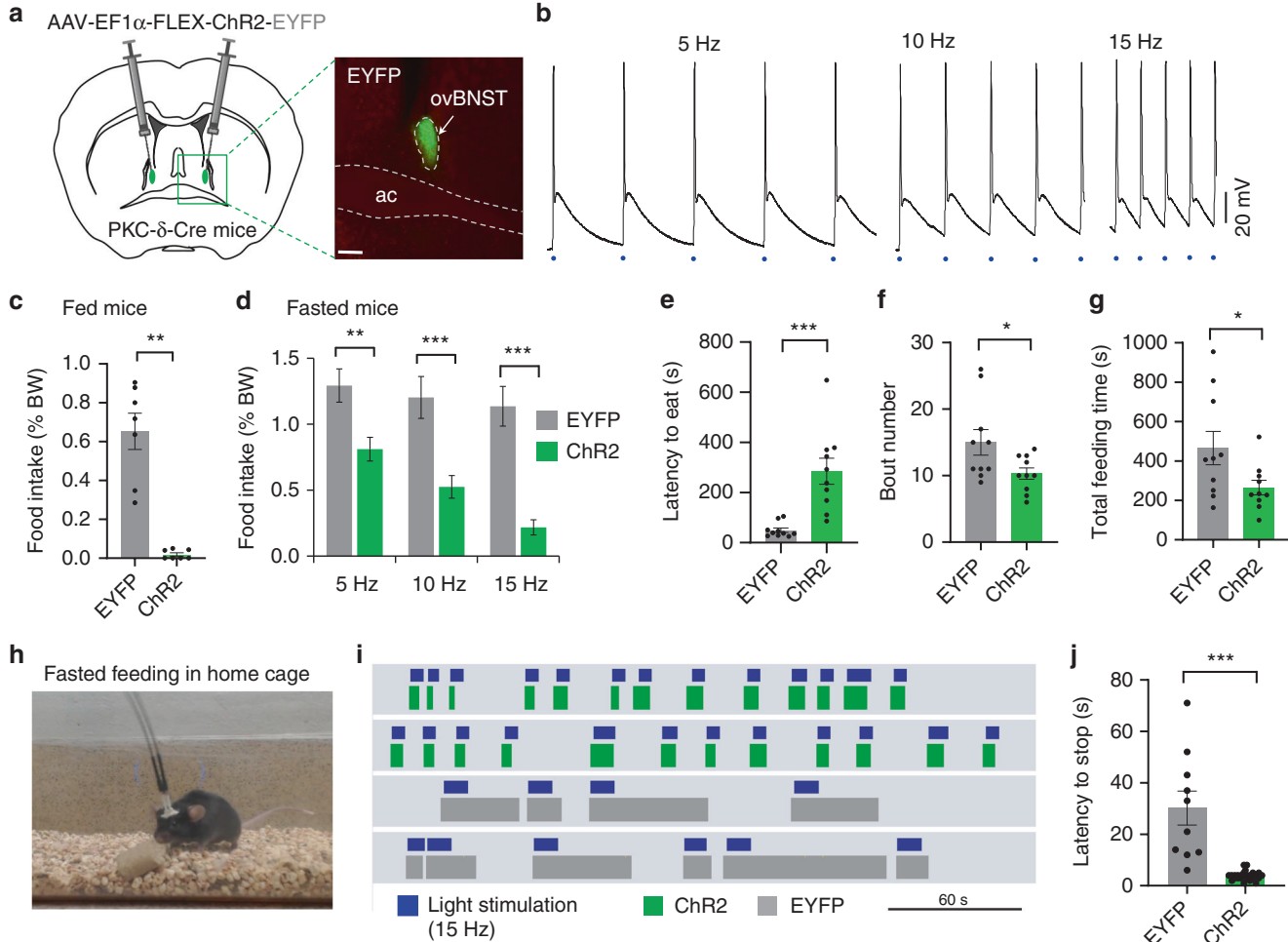

**Fig. 2** Optogenetic activation of ovBNST PKC-δ neurons suppresses feeding. **a** Stereotaxic injection of Cre-dependent AAV to express ChR2-EYFP or EYFP in ovBNST PKC-δ neurons. **b** Brain slice electrophysiological recordings show that ovBNST PKC-δ neurons expressing ChR2 can be activated by light pulses at different frequencies. Blue dots indicate 2-ms light pulses. **c, d** Food intake was suppressed in both fed (**c**) and fasted (**d**) animals when ovBNST PKC-δ neurons were light activated. Unpaired *t*-tests, *t*(12) = 6.74 (**c**), *t*(17) = 3.71 (**d**, 5 Hz), *t*(18) = 4.63 (**d**, 10 Hz), *t*(28) = 6.14, (**d**, 15 Hz), *n* = 7 (**c**) and *n* = 10, 10, and 15 animals for 5 Hz, 10 Hz, and 15 Hz, respectively, in EYFP and ChR2 group (**d**). **e–g** The latency to eat (**e**), bout number (**f**), and the total feeding time (**g**) in fasted animals. Unpaired *t*-tests, *t*(18) = 4.47 (**e**), *t*(18) = 2.23 (**f**), *t*(18) = 2.18 (**g**), *n* = 10 animals in each group. **h–j** Feeding behavior was suppressed when mice were in their home cages. Raster plots (**i**) show that feeding bout duration was decreased in mice expressing ChR2 (green) but not in mice expressing EYFP (gray) in response to 15 Hz light stimulation (blue). Unpaired *t*-test, *t*(33) = 6.40 (**j**), *n* = 10 and 25 for EYFP and ChR2, respectively. Data are mean ± s.e.m. Scale bars, 200 μm. *\*p* < 0.05, *\*\*p* < 0.01, *\*\*\*p* < 0.001. Source data are provided as a separate file

stimulation (Supplementary Fig. 4c). Surprisingly, we did not observe any significant change in the anxiety levels after light activation of the ovBNST PKC-δ neurons (Supplementary Fig. 5a, b). The mobility, determined by the velocity and total distance traveled, was not affected by the optogenetic activation of the ovBNST PKC-δ neurons (Supplementary Fig. 5a, b). Light activation did not produce any conditioned place aversion either (Supplementary Fig. 5c), suggesting activation does not cause significant discomfort or unpleasantness. We also expressed hM3Dq in ovBNST PKC-δ neurons and injected CNO to activate these neurons over an extended time. We found that chemogenetic activation of ovBNST PKC-δ neurons also suppresses food intake in a 2-h feeding session (Supplementary Fig. 4e).

We also tested the feeding when mice were in their home cages, in which the mice were less anxious. Light was delivered within a few seconds following the onset of feeding bout. Our results showed that light activation significantly shortened the average duration of the feeding bouts in mice expressing ChR2 in ovBNST PKC-δ neurons, indicating the feeding was also suppressed in home cages (Fig. 2h–j). Interestingly, although

the feeding behaviors are interrupted by light activation, the animal goes back to feeding quickly after the light is off, suggesting brief activation of the ovBNST PKC-δ neurons impairs neither the feeling of hunger nor the motivation to eat beyond the period of stimulation.

**Silencing of ovBNST PKC-δ neurons increases feeding.** To further determine the role of the ovBNST PKC-δ neurons in feeding, we performed chemogenetic silencing of these neurons in PKC-δ-Cre mice without IL-1β or LPS treatment. We bilaterally injected Cre-dependent AAV encoding hM4Di-mCherry into the ovBNST of PKC-δ-Cre mice for silencing while injected Cre-dependent AAV-mCherry as control. Food intake was measured ~40 min after injecting CNO or saline. We found that chemogenetic silencing of the ovBNST PKC-δ neurons significantly increased the total amount of food intake in both fed and fasted mice (Fig. 3a, b and Supplementary Fig. 6b). Interestingly, chemogenetic silencing of ovBNST PKC-δ neurons significantly increased the bout duration and the total feeding time but not the

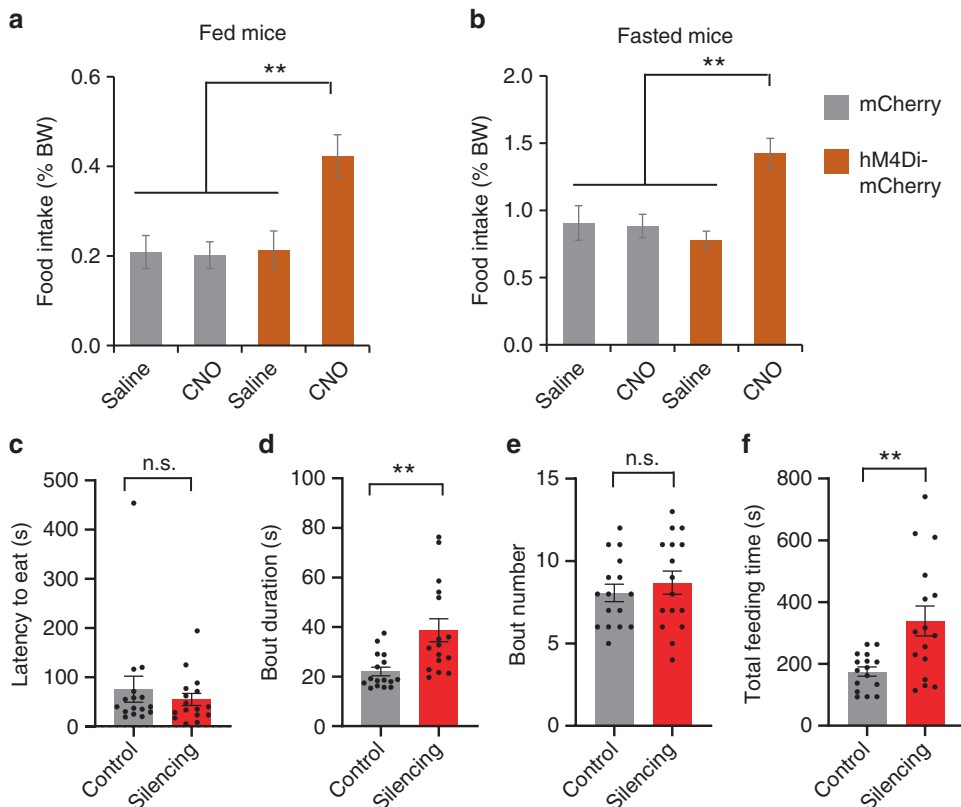

**Fig. 3** Chemogenetic silencing of ovBNST PKC-δ neurons increases feeding. **a**, **b** Food intake was increased when ovBNST PKC-δ neurons were chemogenetically silenced in fed (**a**) and fasted (**b**) animals. Two-way ANOVA with post-hoc Bonferroni's $t$ test, $F_{1,51} = 9.49$, $n = 8$, 12, 16, and 16 animals for mCherry-Saline, mCherry-CNO, hM4Di-saline, and hM4Di-CNO, respectively. **c–f** The latency to eat (**c**), bout duration (**d**), bout number (**e**), and total time that the animals spent in feeding (**f**). Unpaired $t$-test, $t(30) = 0.48$ (**c**), $t(30) = 3.35$ (**d**), $t(30) = 0.71$ (**e**), $t(30) = 3.24$ (**f**, total feeding time). $n = 16$ in each group. Data are mean ± s.e.m. **$p < 0.01$, ***$p < 0.001$. Source data are provided as a separate file

bout number (Fig. 3c–f). Silencing ovBNST PKC-δ neurons increased the amount of food intake in both male and female mice, and we did not observe any significant difference between male and female animals (Supplementary Fig. 6a). Chemogenetic silencing of the ovBNST PKC-δ neurons did not affect the level of anxiety or mobility significantly (Supplementary Fig. 7).

Collectively, these results after activation or silencing of ovBNST PKC-δ neurons suggest that these neurons can not only mediate inflammation-associated anorexia but also bidirectionally modulate feeding behaviors in general.

**ovBNST PKC-δ neurons inhibit neurons in vlBNST**. To understand how the ovBNST PKC-δ neurons regulate feeding, we searched for their downstream targets. After expressing EYFP in ovBNST PKC-δ neurons, the strongest fluorescent nerve terminals were all in BNST subregions, including the anterior lateral BNST and part of anterior medial BNST, both dorsal and ventral to the anterior commissure, and the fusiform BNST (Fig. 4a), neurons in which are mostly GABAergic inhibitory neurons[32,33]. Because all these regions are ventral to the ovBNST and mostly in the lateral part of BNST, we use vlBNST to describe these BNST subregions that are innervated by ovBNST PKC-δ neurons. To test whether neurons in these vlBNST regions receive monosynaptic inputs from ovBNST PKC-δ neurons, we expressed ChR2-EYFP in ovBNST PKC-δ neurons and performed whole-cell patch clamp recordings on vlBNST neurons in brain slices (Fig. 4b; inset). We found that neurons in the vlBNST region with EYFP fluorescent nerve terminals displayed robust inhibitory postsynaptic currents (IPSCs) in response to light activation of

the ovBNST PKC-δ neurons. The IPSCs can be blocked by the GABA_A receptor antagonist picrotoxin (Fig. 4b). The latency of the IPSC is less than 5 ms (Fig. 4c and Supplementary Fig. 8), suggesting that the connection is monosynaptic. There is no significant difference of the IPSC latency and amplitude between male and female animals (Supplementary Fig. 8b, c). We did not observe any excitatory postsynaptic current (EPSC) in vlBNST neurons when ovBNST PKC-δ neurons were light activated, which is consistent with previous studies showing that almost all the neurons in the anterior BNST are GABAergic inhibitory neurons[32,33]. Interestingly, whereas the IPSC latency recorded in the region dorsal to anterior commissure was not different from that in the region ventral to anterior commissure, the IPSC amplitude recorded in the region ventral to anterior commissure is significantly larger than that in the region dorsal to anterior commissure (Fig. 4c), suggesting the ovBNST PKC-δ neurons have a stronger inhibition on the vlBNST neurons in the region ventral to anterior commissure. The firing of vlBNST neurons induced by current injection can be suppressed by light activation of the ovBNST PKC-δ neurons, and the suppression can be blocked by picrotoxin (Fig. 4d). Accordingly, these results demonstrated that ovBNST PKC-δ neurons send monosynaptic inhibition to vlBNST neurons.

**Activation of the neurons in vlBNST increases food intake**. If ovBNST PKC-δ neurons regulate feeding through their inhibitory connections on vlBNST neurons, activation of vlBNST neurons should increase feeding, an effect opposite to the activation of ovBNST PKC-δ neurons. Because vlBNST is close to the ovBNST

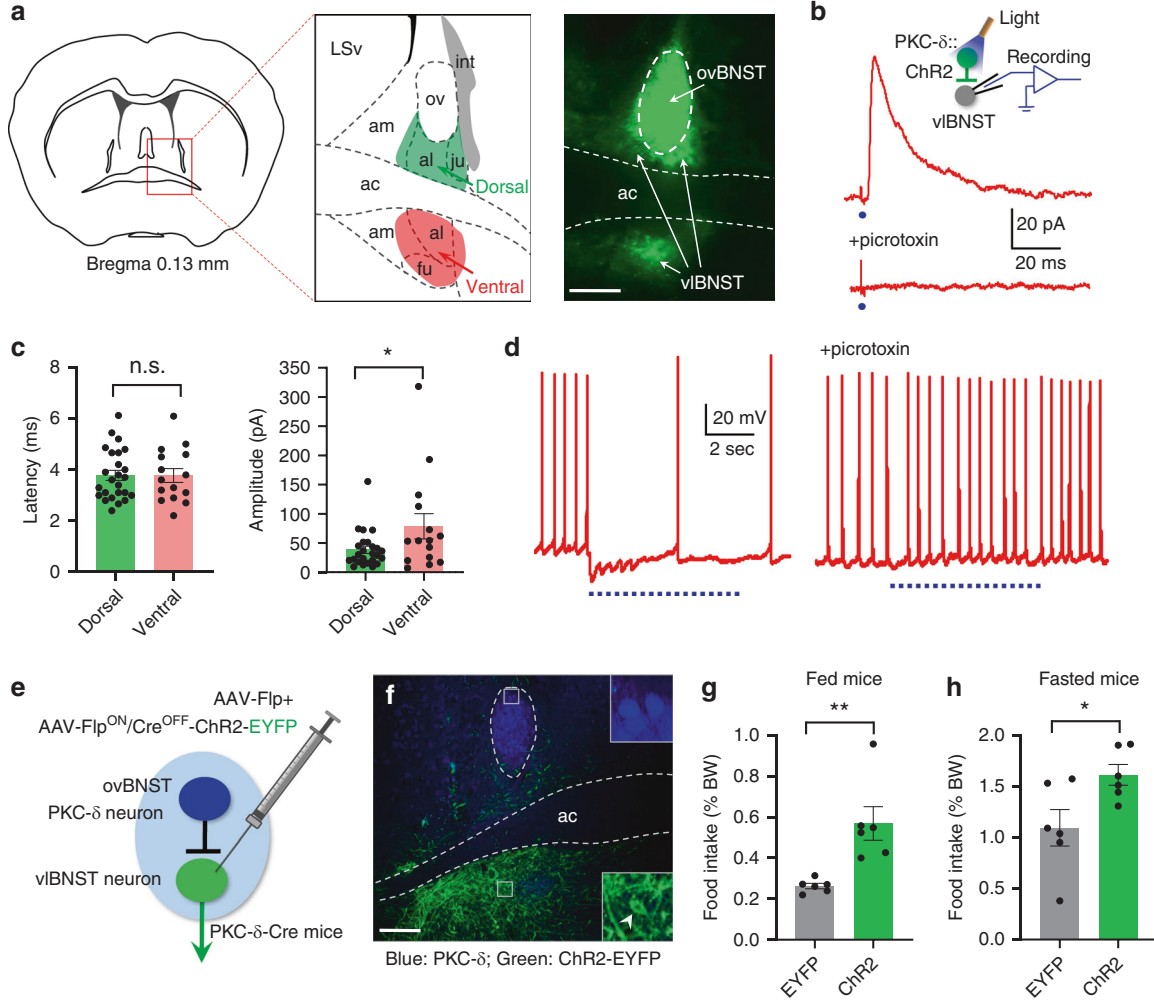

**Fig. 4** Neurons in vlBNST are inhibited by ovBNST PKC-δ neurons and increase feeding when activated. **a** Fluorescent nerve terminals from ovBNST PKC-δ neurons are observed in vlBNST (indicated by arrows) at regions both dorsal and ventral to the ac. Note the image on the right was overexposed to reveal the terminal fluorescence. **b** Whole-cell voltage-clamp recordings on vlBNST neurons show that monosynaptic IPSC can be triggered by light activation of the ovBNST PKC-δ neurons and can be blocked by bath application of picrotoxin. Each blue dot indicates a 2-ms light pulse. **c** The latency and amplitude of the IPSCs from vlBNST neurons in regions dorsal (green) and ventral (red) to ac. Unpaired $t$-test, $t(38) = 0.02$, $p = 0.98$ (latency), $t(38) = 2.2$, $p = 0.035$ (amplitude), $n = 25$ cells from the dorsal region and $n = 15$ cells from the ventral region. **d** Current-clamp recordings on vlBNST neurons show that firing of action potentials can be suppressed by light activation of ovBNST PKC-δ neurons and the suppression can be blocked by picrotoxin. The blue dotted lines indicate the light pulses (15 Hz, 10-ms pulse width). **e** Diagram shows the Cre-out strategy to express ChR2 in vlBNST neurons. **f** ChR2-EYFP was expressed in vlBNST neurons (green, arrowhead indicates a ChR2-EYFP positive vlBNST neuron) but not in ovBNST PKC-δ neurons (blue, immunostaining). **g**, **h** Food intake was increased in fed (**g**) and fasted (**h**) animals when vlBNST neurons were optogenetically activated. Unpaired $t$-test, $t(10) = 3.68$ (**g**), $t(10) = 2.53$ (**h**). $n = 6$ animals in each group. Data are mean ± s.e.m. Scale bars, 200 μm. *$p < 0.05$, **$p < 0.01$. Source data are provided as a separate file

region, we used a Cre-out strategy to express ChR2 in vlBNST neurons while excluding ovBNST PKC-δ neurons. We co-injected AAV-Flp and AAV-Flp$^{ON}$/Cre$^{OFF}$-ChR2[34] into the vlBNST of PKC-δ-Cre mice bilaterally (Fig. 4e) to express ChR2 in neurons that are in the presence of Flp recombinase and in the absence of Cre. We found no PKC-δ neuron in ovBNST expressing ChR2-EYFP (Fig. 4f). Interestingly, even EYFP positive fibers were rarely observed in ovBNST when ChR2-EYFP was expressed in vlBNST, suggesting that vlBNST neurons send very few projections back to the ovBNST. Optogenetic activation of the vlBNST neurons significantly increased the amount of food intake in both fed and fasted mice (Fig. 4g, h), an opposite effect to the activation of ovBNST PKC-δ neurons. Thus, these results support the idea that ovBNST PKC-δ neurons might regulate feeding through their inhibitory connections with vlBNST neurons.

**vlBNST neurons project to LH to promote feeding**. When we expressed EYFP in vlBNST PKC-δ$^−$ neurons, we observed strong fluorescent terminals in LH (Supplementary Fig. 9). To test whether the vlBNST neurons project to LH to regulate food intake, we expressed ChR2 in vlBNST neurons using the same Cre-out strategy and implanted ferrule fibers above LH bilaterally (Fig. 5a). Optogenetic activation of the vlBNST neuron projections in LH significantly increased food intake in both fed and fasted mice (Fig. 5b, c). The level of increase in food intake is dependent on the amount of the light delivered, probably due to the fact that light from the small ferrule fiber (200 μm diameter) cannot effectively cover the large area of the nerve terminals in LH (larger than 1 mm in diameter, Supplementary Fig. 9). We did not observe any significant change in anxiety levels in elevated plus maze or open field tests but a slight

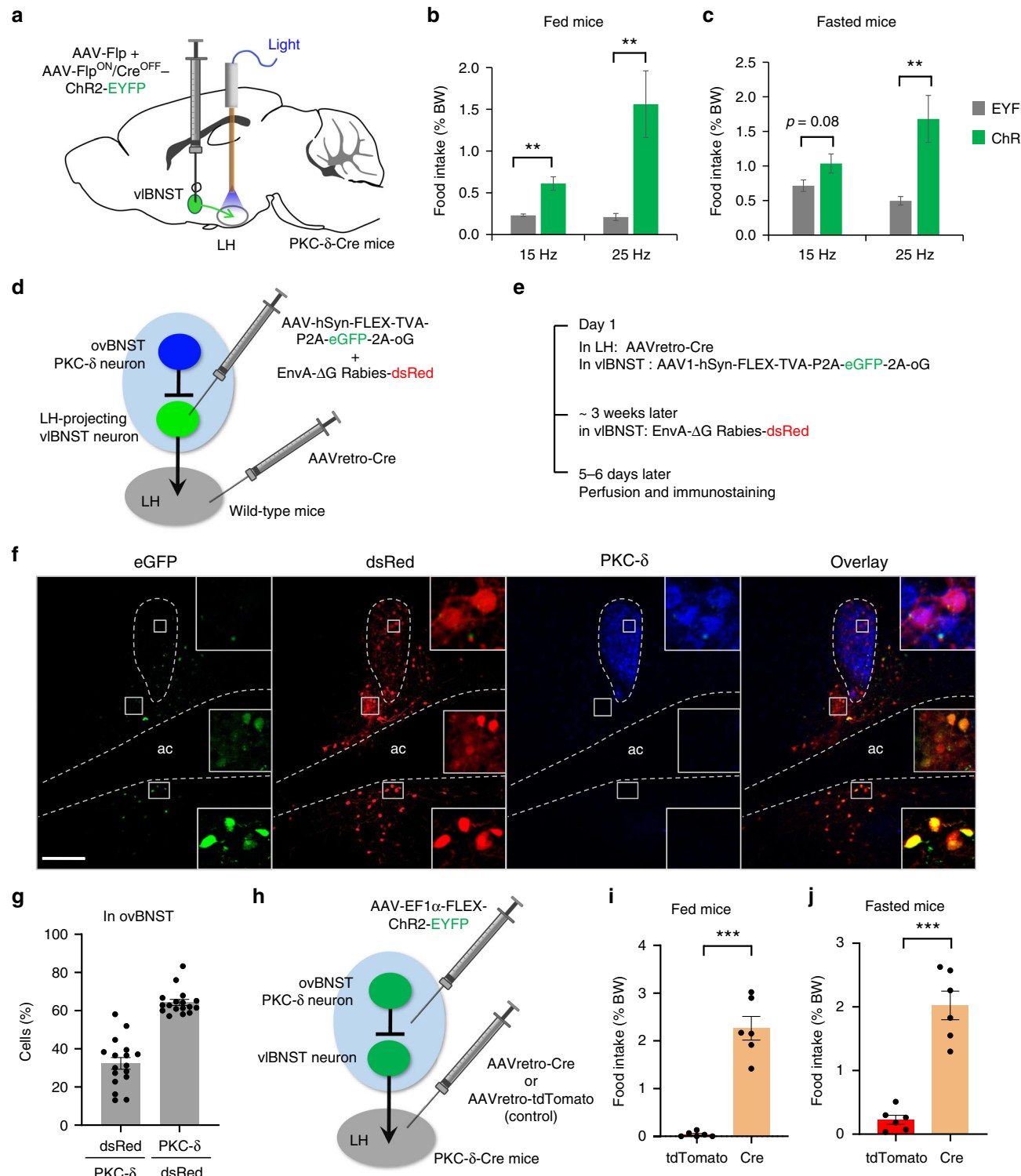

**Fig. 5** LH-projecting vlBNST neurons antagonize feeding suppression induced by ovBNST PKC-δ neuron activation. **a** Diagram shows that Cre-out viruses were injected in vlBNST while ferrule fibers were implanted above LH to activate vlBNST-LH pathway. **b, c** Optogenetic activation of vlBNST-LH pathway increases food intake in both fed (**b**) and fasted (**c**) animals in a frequency-dependent manner. Unpaired *t*-test, *t*(11) = 4.31 (**b**, 15 Hz), *t*(11) = 3.11 (**b**, 25 Hz), *t*(11) = 1.92 (**c**, 15 Hz), *t*(11) = 3.16 (**c**, 25 Hz). *n* = 6 animals for EYFP and 7 for ChR2. **d, e** Diagram (**d**) and virus injection procedure (**e**) to label the upstream neurons that innervate LH-projecting vlBNST neurons. **f** ovBNST neurons that are monosynaptically upstream to LH-projecting vlBNST neurons (green) are labeled by dsRed (red) and overlapping with PKC-δ immunostaining (blue). Note the LH-projecting vlBNST neuron (starter cells, green) are located in regions both dorsal and ventral to act, but not in ovBNST. **g** Quantification of the dsRed cells and PKC-δ neurons in ovBNST. *n* = 17 brain sections from 3 animals. **h** Diagram illustrating the expression of ChR2 in both the ovBNST PKC-δ neurons and LH-projecting vlBNST neurons. The control group was injected with AAVretro-tdTomato in LH, thus only ovBNST PKC-δ neurons will express ChR2. **i, j** Co-activation of the ovBNST PKC-δ neurons and LH-projecting vlBNST neurons increased the food intake in fed (**i**) and fasted (**j**) animals. Unpaired *t*-test, *t*(10) = 8.96 (**i**), *t*(10) = 7.64 (**j**), *n* = 6 animals in each group. Data are mean ± s.e.m. Scale bars, 200 μm. **p < 0.01, ***p < 0.001. Source data are provided as a separate file

increase in the velocity and distance traveled in the open field test upon photoactivation of the vlBNST-LH pathway (Supplementary Fig. 10). Furthermore, chemogenetic silencing of the LH-projecting vlBNST neurons suppresses food intake (Supplementary Fig. 11). These results are consistent with the previous study that showed BNST VGAT neurons project to LH to regulate feeding[21].

**ovBNST PKC-δ neurons innervate LH-projecting vlBNST neurons.** To determine whether ovBNST PKC-δ neurons inhibit LH-projecting vlBNST neurons, we injected retrograde cholera toxin subunit B (CTB, conjugated with Alexa-555) in LH and Cre-dependent ChR2 in ovBNST of the PKC-δ-Cre mice. Then we performed whole-cell patch clamp recording on these CTB back-labeled vlBNST neurons (Supplementary Fig. 12). We found these cells also show monosynaptic IPSCs in response to light activation of the ovBNST PKC-δ neurons (Supplementary Fig. 12c).

To further determine if ovBNST PKC-δ neurons send monosynaptic inputs to LH-projecting vlBNST neurons, we used a Cre-dependent monosynaptic retrograde rabies virus system[35,36] to label the neurons that are upstream to LH-projecting vlBNST neurons. We injected AAVretro-Cre[37] in LH and Cre-dependent AAV-hSyn-FLEX-TVA-P2A-eGFP-2A-oG[36] into the vlBNST of wild-type mice unilaterally (Fig. 5d, e), thus vlBNST neurons that project to LH would be labeled by eGFP and expressing TVA, the receptor for rabies virus. Then we injected G-deleted rabies virus encoding dsRed in vlBNST. In this way, neurons that send monosynaptic inputs to LH-projecting vlBNST neurons would be labeled by dsRed. We found that the LH-projecting BNST neurons (expressing eGFP) were mostly located in vlBNST (both dorsal and ventral to ac), but very few (4.8 ± 0.5%, mean ± s.e.m.) were in ovBNST and positive for PKC-δ staining (Fig. 5f). Approximately 40% of the PKC-δ neurons were expressing dsRed and ~80% of the dsRed neurons in ovBNST were positive for PKC-δ staining (Fig. 5g), suggesting the LH-projecting vlBNST neurons receive inputs predominantly from ovBNST PKC-δ neurons. These findings further confirmed that ovBNST PKC-δ neurons send monosynaptic inputs to LH-projecting vlBNST neurons.

**LH-projecting vlBNST neurons antagonize ovBNST PKC-δ neurons.** The connection from ovBNST PKC-δ neurons to vlBNST neurons, and subsequent projection of vlBNST neurons to LH suggest that these two circuits might function in a row in feeding regulation. We therefore tested whether the LH-projecting vlBNST neurons can antagonize the feeding suppression induced by ovBNST PKC-δ neuron activation. We bilaterally injected AAVretro-Cre in LH and Cre-dependent AAV-ChR2 in both vlBNST and ovBNST regions of the PKC-δ-Cre mice (Fig. 5h), and implanted ferrule fibers above ovBNST. Thus, both ovBNST PKC-δ neurons and LH-projecting vlBNST neurons would express ChR2 and be light activated. In the control group, the mice were injected with AAVretro-tdTomato in LH and Cre-dependent ChR2 in ovBNST and vlBNST. Hence, only ovBNST PKC-δ neurons would express ChR2 and be light activated in the control group animals. Compared to the food intake in the control group, co-activation of the ovBNST PKC-δ neurons and LH-projecting vlBNST neurons strongly increased the amount of food intake in both fed and fasted animals (Fig. 5i, j). These results demonstrated that the LH-projecting vlBNST neurons are downstream to the ovBNST PKC-δ neurons and can overcome the feeding suppression induced by ovBNST PKC-δ neuron activation.

**ovBNST PKC-δ neurons receive inputs from ARC, LPB, and CEA.** To determine the upstream inputs that might regulate ovBNST PKC-δ neurons for feeding, we used the Cre-dependent monosynaptic retrograde rabies system to screen for the brain regions that send inputs to ovBNST PKC-δ neurons (Fig. 6a, b). Multiple brain regions were identified expressing dsRed, the marker for the monosynaptic upstream neurons (Fig. 6c). Importantly, we found dsRed expressed in the ARC, TN, LPB, and CEA neurons, whose role in feeding regulation have been well studied[38–40]. Immunostaining with antibodies against AGRP showed that the dsRed cells in ARC were located in the region positive for AGRP staining (Fig. 6d). However, since the AGRP antibody did not stain cell bodies very well, we cannot make a robust quantification of the percent of dsRed cells in the AGRP population. Around 70% of the dsRed labeled cells in LPB are positive for CGRP staining (Fig. 6e, g). We also found that the dsRed cells in LPB are overlapping with the cells activated by IL-1β (Supplementary Fig. 13a, b), suggesting that the LPB-BNST pathway might regulate inflammation-associated anorexia[12]. CEA is a brain region primarily composed of GABAergic inhibitory neurons[41] which also contain PKC-δ neurons that suppress feeding when activated[17]. Interestingly, most of the dsRed cells in CEA are negative for PKC-δ immuno-staining (Fig. 6f, g). This is consistent with previous studies that showed silencing of CEA PKC-δ negative neurons suppresses feeding[17] while activation might increase feeding[42]. In consistent with a recent study that showed TN neurons project to BNST to regulate feeding[18], dsRed cells were also observed in TN region (Supplementary Fig. 13c).

Altogether, our results demonstrated that ovBNST PKC-δ neurons play an important role in regulating inflammation-associated anorexia, mediate bidirectional control of general feeding. Importantly, the microcircuit from ovBNST PKC-δ neurons to LH-projecting vlBNST neurons might play a central role in feeding regulation by receiving monosynaptic inputs from ARC neurons, LPB CGRP neurons, CEA PKC-δ negative neurons, and other brain regions, and then project to LH to control feeding (Fig. 6h).

## Discussion

The BNST is a heterogeneous brain region with complex cell types and functions, forming diverse connections with many brain regions including those that regulate feeding and energy balance[43]. However, due to its well-established role in sustained fear states, studies on the function of BNST neurons are mostly focused on stress, anxiety, and other emotions[43–46]. Using a genetic marker that labels a specific population of the ovBNST neurons, we discovered a previously undetermined role of BNST neurons in regulating food intake in general and inflammation-associated anorexia in particular.

The BNST microcircuit we identified here might play an important role in coordinating the canonical feeding circuits to regulate food intake. Feeding is a complex behavior comprised of a series of steps including food seeking, initiation, consumption, and termination that are coordinated by neurons distributed across many distinct brain regions[40,47–50], which have been recently summarized as three pillars centered on the three iso-lated brain region of ARC, LH, and LPB[38]. Although some connections of these brain regions have been described, for example, ARC AGRP neurons project to LH and LPB to regulate feeding[38], how neurons in these canonical circuits communicate with each other and coordinate to regulate the complex feeding behavior is still poorly understood.

Recent studies using optogenetics showed that activation of the axon terminals in BNST projected from the GABAergic AGRP neuron in ARC or somatostatin neurons in TN induces feeding

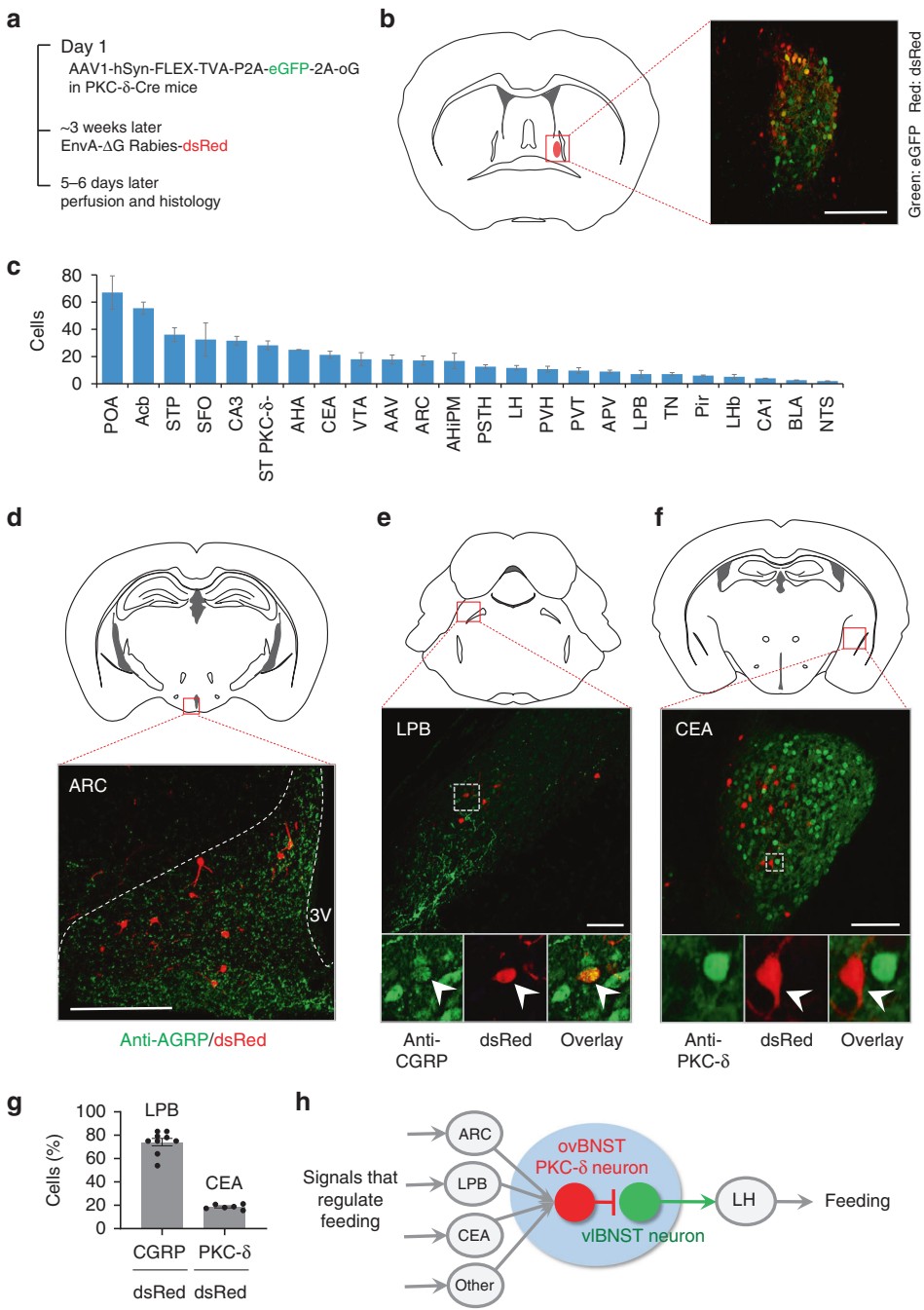

**Fig. 6** ovBNST PKC-δ neurons receive inputs from ARC, LPB, and CEA. **a** Diagram illustrating the procedure of virus injection for monosynaptic retrograde rabies virus tracing. **b** ovBNST PKC-δ neurons expressing eGFP and dsRed as starter cells for retrograde tracing. **c** Major brain regions that send monosynaptic innervation to ovBNST PKC-δ neurons. n = 9 animals. Data are presented as mean ± s.e.m. POA preoptic area, Acb nucleus accumbens, STP BNST posterior, SFO subfornical organ, CA3 hippocampus CA3 region, ST PKC-δ- ovBNST PKC-δ- cells, AHA anterior hypothalamic area, VTA ventral tegmental area, AAV amygdaloid anterior ventral, AHiPM amygdalohippocampal area, posteromedial part, PSTH parasubthalamic hypothalamic nucleus, PVT paraventricular thalamic nucleus, APV amygdaloid posterior ventral, Pir piriform cortex, LHb lateral habenular nucleus, CA1 hippocampal CA1 region, BLA basal lateral amygdala, NTS nucleus solitary tractus. **d–f** Location and representative images show dsRed cells expressed in ARC (**d**), LPB (**e**), and CEA (**f**) and immunostaining of AGRP, CGRP, and PKC-δ, respectively. **g** Quantification of the CGRP, PKC-δ staining in dsRed cells. n = 9 animals (ARC), and 6 animals (CEA). **h** Diagram showing an integrated hierarchy of the brain circuits for feeding regulation. Data are mean ± s.e.m. Scale bars, 200 μm. Source data are provided as a separate file

but activation of the projections from GABAergic BNST neurons to LH also promotes feeding[18,19,21]. These apparently paradoxical results suggested multiple levels of inhibition/disinhibition in the BNST microcircuits. Our finding that ovBNST PKC-δ neurons receive inputs from ARC/TN and inhibit LH-projecting vlBNST neurons for feeding might explain this paradox. However,

whether the ovBNST PKC-δ neurons mediate the feeding induced by neurons in ARC or TN need further experiment to determine.

Our finding that ovBNST PKC-δ neurons receive inputs from LPB CGRP neurons suggests that this pathway might also regulate feeding in conditions of anorexia, which is consistent with previous studies that show activation of LPB CGRP neurons

suppresses feeding[12,14]. It has been demonstrated that LPB CGRP neurons send excitatory projections to CEA PKC-δ neurons, which suppress feeding by inhibiting the GABAergic CEA PKC-δ negative neurons[17]. This is also consistent with our result that CEA PKC-δ negative neuron is the dominant subpopulation in CEA innervating ovBNST PKC-δ neurons.

Thus, the BNST microcircuit we identified here not only provides a connection between the three pillars, but also suggests a mechanism that integrates the three pillar circuits for feeding regulation (Fig. 6h). How the different nodes of this circuits structure coordinate to regulate feeding and their dynamics during the complex behaviors of feeding will be exciting areas for future research. It should be noted that the BNST neural circuits for feeding might be more complicated than this scheme. Both ovBNST PKC-δ neurons and vlBNST neurons receive diverse inputs and project to various brain regions. Whether these pathways also regulate feeding remains to be determined. Moreover, BNST and CEA share many similar anatomical connections and functions[43]. CEA PKC-δ neurons also suppress food intake when activated and send inhibitory projections to vlBNST[17]. Future studies will have to determine whether the CEA PKC-δ neurons and ovBNST PKC-δ neurons are synergistic or parallel and redundant in feeding regulation.

Although numerous studies have suggested that the neural circuits for feeding might play a role in regulating inflammation-associated anorexia[1–3,51], the exact brain region and type of neurons responsible for initiating the anorexia are still unclear. Intra-4th ventricle delivery of the glucagon-like peptide-1 (GLP-1) antagonist or metabotropic glutamate antagonist attenuates the anorexia after LPS treatment[52,53], indicating that neurons in the brain stem might regulate inflammation-associate anorexia. Indeed, chemogenetic silencing of LPB CGRP neurons can partially prevent the LPS-induced anorexia[14] and continuous inactivation of the CGRP neurons prevents cancer-induced anorexia, a chronic condition associated with inflammatory processes[12]. Disruption of serotonin receptor or inducible nitric oxide synthase by drug infusion also attenuates IL-1β or LPS-induced anorexia[54–56], but the exact responsible brain region is unknown. Peripheral or central administration of IL-1β or LPS induces c-Fos expression in many brain regions involved in feeding and energy homeostasis, including brain stem LPB, CEA, BNST, ARC, and paraventricular hypothalamus (PVN), but reduces c-Fos expression in LH neurons (for example, refs. [9–13]). Surprisingly, very few of these brain regions have been demonstrated to regulate inflammation-associated anorexia. Rather, many studies suggested a negative role of these brain regions. Lesion of the ARC fails to reverse the anorexia induced by IL-1β[13]. Activation of ARC AGRP neurons potently induces feeding but cannot restore LPS-induced anorexia[15]. Instead, LPS completely suppresses AGRP neuron-induced food intake[16]. Silencing the CEA PKC-δ neurons restores the feeding suppression induced by satiety or bitter tastants but has no effect on LPS-induced anorexia[17]. Thus, our finding that ovBNST PKC-δ neurons are preferentially activated by IL-1β or LPS and silencing of these neurons significantly attenuates the anorexia induced by IL-1β or LPS is important in identifying a unique brain region and a specific population of neurons that can mediate inflammation-associated anorexia. It should be noted that our c-Fos mapping shows only ~20% of the PKC-δ neurons are activated by IL-1β or LPS. This could be due to the low sensitivity of c-Fos expression in detecting activated neurons, or that some ovBNST PKC-δ neurons may play a different role. However, the diverse brain regions that we identified to be upstream to the ovBNST PKC-δ neurons and the downstream LH-projecting vlBNST neurons further offer insights to determine the brain regions responsible for inflammation-associated anorexia. Future studies to examine

the role of these neurons in inflammation-associated anorexia are therefore warranted.

Together, our study defines a microcircuit within BNST that plays an important role in regulating inflammation-associated anorexia and feeding behaviors in general. Specific neurons and small neural circuits in the BNST, therefore could serve as potential therapeutic targets for inflammation-associated anorexia and other feeding-related diseases such as eating disorders and obesity.

## Methods

**Mice**. To ensure that the mice we used in this project have a consistent genetic background, we crossed the PKC-δ-Cre mice with the wild-type C57BL/6 mice from the Charles River Laboratory (a background used in our previous study[17]) for at least 5–6 generations. The genotype of transgenic PKC-δ-Cre mice offspring is identified by PCR on genomic tail DNA. Both wild-type and PKC-δ-Cre offspring were used in this study. Survival surgery was performed when mice are 2–3 months old and behavioral tests were performed when mice are 3–5 months old. All mice were housed on a 12-h light (7 am)/dark (7 pm) cycle with ad libitum access to water and rodent chow unless placed on a food restriction schedule for fasted feeding experiments. All behavioral experiments or tissue collection for ex vivo slice electrophysiology were performed during the light cycle. Because we did not observe any difference between male and female mice in our experiments (e.g., Supplementary Figs. 2, 3, 5, and 7b, c), unless indicated, we usually analyzed the results by combining approximately the same number of male and female mice throughout the study.

All animal care and experimental procedures complied with all relevant ethical regulations, were strictly conducted according to the guidelines of US National Institutes of Health for animal research and were approved by the Institutional Animal Care and Use Committee (IACUC) at the University of Arizona.

**Virus and tracer**. AAV2-EF1a-DIO-EYFP-WPRE-pA, AAV2-EF1a-DIO-hChR2 (H134R)-EYFP-WPRE-pA, AAV5-EF1a-mCherry-IRES-Flpo-WPRE, and AAV5-hSyn-Cre^OFF/Flp^ON-hChR2(H134R)-EYFP-WPRE were generated by Dr. Karl Deisseroth's lab at Stanford University. AAV5-hSyn-DIO-mcherry and AAV5-hSyn-DIO-hM4Di-mCherry were generated by Dr. Bryan Roth's lab at the University of North Carolina (UNC). AAV2retro-Cre and AAV2retro-tdTomato were generated by Dr. Ed Boyden's lab at MIT. All these viral constructs were deposited and packaged into viral vectors in the UNC Viral Vector Core or Addgene. All the AAV and AAV2retro viruses had titers of $1–6 \times 10^{12}$ genome copies per ml. Construct validity and correct targeting to the brain nucleus of interest are confirmed through post-mortem processing of brain tissue sections in multiple sets of mice or electrophysiological recordings on live brain slices. Viruses were usually injected in mice at 2–3 months old. With stereotaxic injection, the viruses were usually expressed in more than 90% of the ovBNST PKC-δ neurons. Behavioral experiments were performed 4–8 weeks after virus injection. To minimize variation in environmental differences across days, mice behaviors, etc., the control virus and experimental virus surgery were performed in the same time window by the same investigator.

For rabies virus tracing, AAV1-hSyn-FLEX-TVA-P2A-eGFP-2A-oG virus and EnvA G-deleted Rabies-dsRed virus were generated by Dr. Edward Callaway's lab and produced at the Gene Transfer, Targeting and Therapeutics Facility of the Salk Institute for Biological Studies in La Jolla. EnvA G-deleted Rabies-dsRed was injected 3 weeks after the injection of AAV1-hSyn-FLEX-TVA-P2A-eGFP-2A-oG virus at the same coordinates (see "Surgery" below).

Cholera Toxin Subunit B (CTB 555, Alexa Fluor™ 555 Conjugate, Invitrogen, c34776) was used as a retrograde tracer to fluorescently label LH-projecting BNST neurons during ex vivo slice electrophysiology recordings. CTB 555 was diluted to 0.1% (w/v) in sterile phosphate-buffered saline (PBS), aliquoted and stored at −20 °C before using.

**Stereotaxic animal surgery**. All mouse survival surgeries were performed using aseptic techniques. Briefly, mice 2–3 months old were deeply anesthetized with 5% isoflurane in oxygen and kept at 1–1.5% isoflurane during surgery. Surgery was performed with a stereotaxic frame (Model 1900 Stereotaxic Alignment System, Kopf Instruments). An incision was made down the midline of the scalp and a craniotomy was performed above the target regions. Viruses or the fluorescent tracer CTB 555 were microinfused through a pulled-glass micropipette with 20–50 μm tip outer diameter connected with a Nanoliter Injector (Nanoliter 2010, World Precision Instruments) at a rate of 20–50 nl min⁻¹. After injection, the micropipette was left in place for 5 min to allow for diffusion of the liquid before the pipette was slowly withdrawn. Injection volumes into the ovBNST, vlBNST, and LH were 200, 150, and 150 nl, respectively. Virus or tracer was injected bilaterally for behavioral studies and slice electrophysiology. Animal for rabies virus tracing experiments received injections unilaterally. Injection coordinates (in mm) relative to midline and bregma: ovBNST (±1.10, +0.15, −4.30), vlBNST (±1.10, +0.15, −4.70), LH (±1.10, −1.40, −4.90). For behavioral tests, optical

ferrule fibers were implanted bilaterally ~0.5 mm above the injection coordinates. After ferrule fiber implantation, dental cement (C&B Metabond) was used to secure the fiber to the skull. For postoperative care, mice were injected intraperitoneally with ketoprofen (5 mg/kg) daily for 3 days. At least 4 weeks post surgery were allowed for mouse recovery and viral expression before the behavioral assays.

**Pharmacology.** Clozapine-N-oxide (CNO) (Enzo Life Science-Biomol, BML-NS105-0005) was freshly dissolved in injection saline (0.9% NaCl) to a concentration of 1 mg per ml and intraperitoneally injected at 5 mg per kg for hM4Di silencing. The 1 mg per ml CNO was further dissolved to 0.1 mg per ml and injected at 0.5 mg per kg for hM3Dq activation. Other compounds used for intraperitoneal injection were IL-1β (5 μg per kg[9], recombinant human interleukin-1β, BD Biosciences, No. 554602), LPS (0.2 mg per kg[55], Sigma, L4516-1MG), TNFα (100 μg per kg[57], BD Biosciences, No. 554618), CCK (5 μg per kg[17], Tocris, #1166). IL-1β, LPS, TNFα, and CCK were freshly dissolved in saline to concentration of 2.5 μg per ml, 0.1 mg per ml, 20 μg per ml, 0.1 μg per ml, respectively. LiCl (150 mg per kg[17], Sigma) was prepared in 150 mM with $dH_2O$. Behavioral tests were usually performed 40 min after CNO injection and 50–60 min after the injection of IL-1β or LPS. Saline was injected as vehicle control.

**In vivo optogenetics.** Blue laser (Shanghai DreamLaser: 473 nm, 100 mW) was used to deliver light stimulation. An Accupulser Signal Generator (World Precision Instruments, SYS-A310) was used to control the frequency and pulse width of the laser light. Light was delivered to the brain through an optic fiber (200 μm diameter, NA 0.22, Doric Lenses) connected with the implanted ferrule fiber by a zirconium sleeve. The light power in the brain regions 0.5 mm below the fiber tip was calibrated as previously described[58]. The calibrated light power density (0.5 mm below the fiber tip) used in light activation experiment was ~5 mW per mm². 5, 10, 15, and 25 Hz, 10-ms (pulse width) light pulse trains were used in different optogenetic activation experiments.

**Feeding assays.** One day before the first feeding test, mice were transferred into an empty testing cage in the behavioral testing room to habituate for at least 20 min. For the 24-h fasted feeding test, mice were food-deprived, with water provided ad libitum, 1 day before test. Before the experiment, mice were briefly anesthetized with isoflurane and coupled with optic fibers. At least 25 min after recovery in the same behavioral testing room, mice were introduced into a clean empty testing cage with a pre-weighed regular food pellet, and allowed to feed for 20 min. The body weight of the mice before test, weight of food pellet before and after test, including the food debris left in the cage floor after test, were measured to calculate the net food intake. For the fed feeding test, mice were not food deprived before testing, and allowed to feed for 30 min. Unless indicated, all the feeding durations are 20 min for fasted state and 30 min for fed states. For optogenetic experiments, the light was delivered just after the mice were introduced into the testing cage. After each test, mice were returned to home cage with ad libitum access to water and rodent chow. For the home cage feeding test, mice were food deprived for 24 h. A single food pellet was placed in the home cage at the beginning of the test and the animal was allowed to eat for 10 min. Activation light (473 nm) was triggered 1–2 s after each feeding behavior began. 15 Hz, 10-ms light pulses were delivered for 10 s or stopped 1–2 s after the cessation of each feeding bout. The feeding behavior was videotaped and manually analyzed with a MATLAB based in-house behavioral annotation script. For pharmacogenetic experiments, CNO or vehicle was injected 40 min before the feeding test. For experiments that require injection of CNO or vehicle multiple times, CNO or vehicle were counterbalanced and at least 3 days were allowed between the injections. All feeding tests were performed between 2 pm and 7 pm.

**Elevated plus maze.** A standard elevated plus maze (40 cm above the floor) with two opposing open arms (30 × 5 × 15 cm) and two opposing closed arms (30 × 5 × 15 cm) was used to measure anxiety level. Mice were placed into the center of the elevated plus maze and their position was tracked with Ethovision offline. For optogenetic tests, mice were allowed to explore during a 6-min session. The 6 min session was divided into three 2-min periods: one without any light stimulation, one with light stimulation (473 nm, 10 ms pulse, 15 Hz) and a final one without stimulation. All behaviors were videotaped and analyzed offline with Ethovision (XT 10.0, Noldus Information Technology).

**Open field.** A white square box (50 × 50 × 30 cm, a 25 × 25 cm square center was defined as "center" in analysis) was used as open field box. Mice were placed individually in the center of the box, and their behavior was tracked for 9 min in optogenetic tests with 3 min of light stimulation (473 nm, 10 ms pulse, 15 Hz) applied 3 min after the start. All the behaviors were videotaped and analyzed offline with Ethovision.

**Conditioned place preference test.** The conditioned place preference test was performed with a three-chamber system (40 × 25 × 15 cm), in which the walls of chamber A and chamber B differ in appearance and texture, while the center chamber is a neutral enclosure. Mice were allowed to explore all three chambers for

10 min on day 1 for preconditioning. On conditioning trials, mice were restricted in one side of the chamber for 10 min with continuous light stimulation. Then they were restricted to the other side for 10 min without light stimulation. Conditioning trials occurred for 2 days on days 2 and 3. On day 4 mice were allowed to explore all three chambers for 10 min. Their behavior was recorded and analyzed offline with Ethovision.

**Immunohistochemistry and histology.** All mice used for behavioral tests and anatomical experiments were deeply anesthetized with isoflurane and ketamine/xylazine, then perfused and checked for virus expression and optical fiber positioning. For immunofluorescent staining, mice were transcardially perfused with 20-ml PBS followed by 20-ml of 4% paraformaldehyde in PBS. Brains were removed and post-fixed in 4% paraformaldehyde overnight before being rinsed twice with PBS. The brains were sectioned with a vibratome (Leica, VT1000S) at 45 μm thickness. Sections were stained with primary antibody at 4 °C overnight, in a blocking solution containing 5% donkey serum and 0.5% Triton X-100. After 3 × 10 min wash in PBS, standard Alexa Fluor secondary antibodies (Jackson Immuno Research Inc., 1:500) were added at room temperature for 1–2 h. Sections were then washed 3 × 10 min in PBS and mounted on glass slides and coverslipped using Vectashield Fluo Gel (H1500 with DAPI or H1000 without DAPI) and viewed under a ZEISS AxioZoom V16 Fluorescent Microscope with Apotome 2 Structured Illumination Module for optical sectioning. Primary antibodies used: rabbit anti-PKC-δ (Abcam, ab182126, 1:1000), goat anti-CGRP (Abcam, ab36001, 1:200), rabbit anti-AGRP (Phoenix Pharmaceuticals, Inc., No. 01765-3, 1:200), and goat anti-c-Fos (Santa Cruz Biotech, sc-52-G, 1:500).

*PKC-δ and c-Fos staining analysis*: Mice were perfused 75–90 min after the injection of IL-1β or LPS and brains were sectioned, stained with anti-PKC-δ and anti-c-Fos, and imaged with the procedure described above. Typically, 3–5 brain sections that include anterior, middle, and posterior ovBNST regions were analyzed and averaged per animal. The number of brains was indicated in the legends of individual figures. Florescence-images for colocalization and quantification were performed with the cell counter plug-in in FIJI (ImageJ). Cells were scored as either PKC-δ+ only, c-Fos+ only, PKC-δ+, and c-Fos+.

*EnvA G-deleted-dsRed Rabies cell counting*: 21 days after the injection of AAV1-hSyn-FLEX-TVA-P2A-eGFP-2A-oG and 5–6 days after EnvA G-deleted Rabies-dsRed injection, mice were perfused, and brains were sectioned, immunostained and mounted, and imaged as described above. Brain slices were immunostained with anti-PKC-δ (in BNST and CEA), anti-AGRP (in ARC), and anti-CGRP (in LPB) antibodies. Each dsRed+ cell was assigned to a specific anatomical structure by using the Mouse Brain in Stereotaxic Coordinates (coronal plates 20–80)[59]. The number of fluorescent cells were counted using the cell counter plug-in in FIJI (ImageJ).

**Electrophysiological slice recordings.** Mouse brain slice electrophysiology recording was performed as described[17]. In brief, mouse brain coronal sections were sectioned at 250 μm thickness with a vibratome (Leica, VT1000S) on ice, using the artificial cerebrospinal fluid (ACSF) containing 126 mM NaCl, 1.6 mM KCl, 1.2 mM $NaH_2PO_4$, 1.2 mM $MgCl_2$, 2.4 mM $CaCl_2$, 18 mM $NaHCO_3$, 11 mM glucose (oxygenated with carbogen (95% $O_2$ balanced with $CO_2$) for at least 15 min before use). The brain sections after cutting were immediately transferred to NMDG-HEPES recovery solution (93 mM NMDG, 2.5 mM KCl, 1.2 mM $NaH_2PO_4$, 30 mM $NaHCO_3$, 20 mM HEPES, 25 mM Glucose, 5 mM sodium ascorbate, 2 mM thiourea, 3 mM sodium pyruvate, 10 mM $MgSO_4$, 0.5 mM $CaCl_2$, 300–310 mOsm, titrated with 10 N HCl to adjust pH to 7.3–7.4) for recovery, 15 min at 32–34 °C. Brain slices were then transferred to ACSF at room temperature and recordings were performed 1 h later in a rig equipped with a fluorescence microscope (Olympus BX51), MultiClamp 700B and Digidata 1550A1 (Molecular Devices). The patch pipettes with a resistance of 5–10 MΩ were pulled with P-97 Sutter micropipette puller and filled with an intracellular solution (135 mM potassium gluconate, 5 mM EGTA, 0.5 mM $CaCl_2$, 2 mM $MgCl_2$, 10 mM HEPES, 2 mM MgATP, and 0.1 mM GTP, pH 7.3–7.4, 290–300 mOsm). Recording data were sampled at 10 kHz, filtered at 3 kHz and analyzed with pCLAMP10. Classification of BNST neurons based on their firing pattern in response to the current injections was described as before[28,29]. For the optogenetic stimulation, either a laser (Shanghai DreamLaser, 473 nm, 50 mW) or a Blue LED light source (Doric Lenses) was used to deliver light pulses (0.1–5 mW per mm² at the tip). 2 ms light pulses were used to trigger action potentials in cells expressing ChR2 and induce IPSC in cells postsynaptic to ChR2-expressing cells. To match the ovBNST PKC-δ neurons manipulated in the behavioral experiments, the PKC-δ neuron electrophysiological properties were characterized in ovBNST neurons that show no-delay action potentials or ChR2-currents in response to light pulses after virus expression of ChR2.

**Quantification and statistical analysis.** Data represent mean ± s.e.m or median as indicated. Unpaired Student's *t*-test was used to compare two groups and one-way ANOVA with post-hoc Bonferroni's *t*-test was used to compare three or more groups with one variable. Two-way ANOVA was used for data with more than one independent variables. A *p* value smaller than 0.05 was considered significant. Data were analyzed with GraphPad Prism Software.

**Reporting summary**. Further information on research design is available in the Nature Research Reporting Summary linked to this article.

## Data availability

The source data for each figure are provided as a Source Data file as indicated in figure legends. All the data that support the findings of this study are available from the corresponding author upon reasonable request.

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

## Acknowledgements

The authors thank W. Haubensak and D. Anderson for PKC-δ-Cre mice; K. Deisseroth, E. Boyden, B. Roth, and E. Callaway for viruses; K. Gothard, F. Porreca, X. Xu, and K. Zinsmaier for comments and critical reading of the manuscript. This work was supported in part by a NARSAD Young Investigator Award from the Brain & Behavior Research Foundation and Bio5 Institute at the University of Arizona (to H.C.).

## Author contributions

H.C. conceived the project. Y.W. and H.C. designed the experiments. Y.W. performed all the behavior and histology experiments, J.K. performed the electrophysiology experiments, M.B.S. wrote the MATLAB software for behavioral annotation, T.S.C. helped the behavioral annotation, C.F. managed the mice colony, characterization, and genotyping. Y.W. and H.C. analyzed the data and wrote the manuscript with help from M.B.S. and T.S.C.

## Additional information

**Competing interests:** The authors declare no competing interests.

