## [Peer Review File · Nature Communications]

Reviewers' comments:

Reviewer #1 (Remarks to the Author):

This study by Wang et al. maps a novel feeding circuit that contributes to the loss of appetite associated with inflammation. There is much interest in the neural underpinnings of anorexia specifically and appetite regulation in general, due to the cost of obesity and feeding related disorders to society. Many feeding centers in the brain, such as hypothalamic AgRP neurons, profoundly regulate appetite but cannot overcome the loss of appetite induced by lipopolysaccharide (LPS) injections.

This manuscript demonstrates that PKC- δ neurons in the ovBNST regulate inflammation-induced anorexia and exert bidirectional control over feeding behavior more generally (chemogenetic inhibition promotes food intake, while photo-activation decreases it) and. The authors do a great job mapping the inputs and outputs of these cells and convincingly show that the feeding effects are mediated through monosynaptic inhibitory inputs to vBNST neurons that project to the lateral hypothalamus. Altogether, the work beautifully describes a novel circuit that mediates inflammation-related suppression of feeding.

The paper is well written, the stats are clear, and the figures are easy to understand. The authors' claims are very well supported, and the appropriate literature is cited. The work is straightforward, but will be certainly be of great interest to researchers interested in feeding behavior. I have no major concerns with manuscript. I commend the authors for completing a great project and only have a few suggestions.

Minor concerns:

1. The involvement of this circuit in loss of appetite is a key strength of the paper. To bolster this relationship, I wish the authors had tested if inhibiting vBNST to LH neurons reduces feeding. The photostimulation experiments done on this pathway are interesting, but not very physiological (due to the synchronicity of photostimulation-induced activity).
2. If the authors' tried using excitatory DREADDs to alter feeding behavior, I encourage them to add that data to the manuscript. Since the inhibitory DREADD experiments and excitatory opto experiments both worked so well, it would be interesting to know if excitatory DREADDs were any less effective in altering feeding behavior. The two approaches cause very different types of elevations in spiking activity.
3. It would be nice to see if the vBNST neuron manipulations altered behavior in a general manner (e.g., locomotor activity and anxiety), as was tested with the ovBNST neuron manipulations.
4. Supplemental figure 1 should be highlighted in the results section. Panel K is particularly interesting and deserves a mention in the text. Also, do you know what percent of ovBNST neurons are PKC- δ positive?
5. Considering that "act" is already a word, it might not be the best acronym for anterior commissure. I am not sure that an acronym is even necessary here.

Reviewer #2 (Remarks to the Author):

In this manuscript, Wang and colleagues have identified a new BNST microcircuit involving PKC-delta expressing neurons, which plays a role in inflammation-associated anorexia and in feeding behavior. The study is original and well performed and the manuscript well written. Statistical

analysis is appropriate.

Comments arising from the reading of the manuscript are further detailed below.

Major :

3. Why did the authors think that the PKCdelta subpopulation in ovBNST might be relevant to characterize? The initial hypothesis that guided the authors towards this specific subpopulation is unclear and needs to be addressed in the introduction.

1. Is the activation of PKC-delta neurons in ovBNST specific to IL1beta and LPS ? What about other important inflammatory signals such as TNFalpha or IL-6? It would be relevant to understand whether these cells respond only to certain inflammatory stimuli, or not.

2. Related to the point above: do the authors have any evidence that this sub-group of cells is activated in cancer-bearing or other mice models of inflammation-associated anorexia?

3. The authors report in figure 1D that only 20% of PKC-delta cells are activated in response to inflammatory stimuli. This represents quite a small group of cells after all and this point should be discussed in the discussion section.

Minor :

4. What was the success rate of expression of Chr2 and hM4Di in PKC-delta neurons ? was this verified? This information should be included in the methods and supplementary figures.

2. In the methods, please specify if C57Bl/6 used were Bl6J or Bl6N.

3. In the methods, please give more details on how the double immunofluorescent staining for PKC-delta and c-fos was actually performed. Please also define how many sections per brain and how many brains per condition were evaluated.

Reviewer #3 (Remarks to the Author):

Wang et al., reported that the oval region of the BNST mediated inflammation-associated anorexia by using chemo/optogenetic approaches, and demonstrated a local circuit in the BNST. This study covered both BNST and LH, which have been previously extensively studied in the fields of feeding behavior. The neural mechanisms of inflammation-associated anorexia is relatively unclear, which was interesting. However, there were several major problems which largely reduced my enthusiasm. For example:

1. In the figure 1a, the adjacent areas were too dim to recognize, which are important to clarify the studied brain area (ovBNST);

2. One big concern was the number of inflammation-related Fos-positive neurons in Fig. 1c-d was too small (less than 20). How could this small number of neurons affect behaviors, for example, feeding behavior in this study?. This was the "core" part of this study, but it was meaningless.

3. Another big concern was why only 20 min food intake was recorded and analyzed? If not affecting long term food intake (ie. hours), it was meaningful either. Moreover, this reviewer was not comfortable with analyzing feeding behavior by using food intake per body weight. The amount of food intake should be used to perform feeding assays.

4. Following #3, 24 hr food deprivation would stress animals, so physiological feeding assays (ie. feeding in later afternoon or evening or night feeding should be repeated by using similar manipulations.

5. In Fig. 2, feeding assays should be performed on same mice before, during and after photostimulation. Otherwise, individual variations in feeding would bring "off-target" effects.

Reviewer #4 (Remarks to the Author):

The manuscript by Wang et al. describes a new BNST microcircuit that suppresses food uptake in response to satiety and inflammatory signals. Using c-fos immunoreactivity as a marker for neuronal activity, the authors identify in the oval region of the BNST a subpopulation of protein kinase C-delta expressing neurons, which become activated by inflammatory signals such as IL-3beta or LPS. Chemogenetic silencing of ovBNST-PKCdelta neurons attenuated the anorexia induced by IL-3beta or LPS, and increased food intake of fed and fasted mice. Optogenetic activation of ovBNST-PKCdelta neurons produced the opposite effect. ovBNST-PKCdelta neurons send inhibitory projections to neurons in a ventral-lateral subregion of the BNST, and these vBNST neurons project to the lateral hypothalamus to promote feeding. These LH-projectors are downstream of the ovBNST-PKCdelta neurons, because they can overcome the feeding block induced by ovBNST-PKCdelta neuron activation. Finally, the authors use monosynaptic Rabies tracing to map the inputs of ovBNST-PKCdelta neurons.

I find this study interesting and the data largely convincing. The authors used state-of-the-art circuit mapping tools to discover a novel role of BNST neurons in regulating feeding. I only have minor points, which the authors could easily address without performing additional experiments.

1. It is unusual that food consumption is normalized to the body weight of the animal. Considering that all the results were normalized in that way, I am wondering whether the authors could show that food consumption actually correlates with BW in their experimental groups.
2. In figure 2i, it is interesting to note that although the feeding bouts are shorter because they are interrupted by light activation, they are more numerous. This suggests that while activation of ovBNST-PKCdelta cells suppresses food intake, it does not appear to impair the hunger feeling nor the motivation to consume food. The authors may want to comment on this in the manuscript.
3. In figure 5b,c, the legend does not provide information on the duration of the experiment.
4. In figure 6g, the fraction of AGRP+ dsRed+ cells was not quantified. The authors should quantify or state why it was impossible to do so.

Response to reviewers: Wang et al.

We would like to thank the reviewers for their excellent and insightful comments. We have performed new experiments and improved the manuscript in response to reviewers' comments. Please find below our response detailing these changes.

Reviewer #1

This study by Wang et al. maps a novel feeding circuit that contributes to the loss of appetite associated with inflammation. There is much interest in the neural underpinnings of anorexia specifically and appetite regulation in general, due to the cost of obesity and feeding related disorders to society. Many feeding centers in the brain, such as hypothalamic AgRP neurons, profoundly regulate appetite but cannot overcome the loss of appetite induced by lipopolysaccharide (LPS) injections.

This manuscript demonstrates that PKC- δ neurons in the ovBNST regulate inflammation-induced anorexia and exert bidirectional control over feeding behavior more generally (chemogenetic inhibition promotes food intake, while photo-activation decreases it) and. The authors do a great job mapping the inputs and outputs of these cells and convincingly show that the feeding effects are mediated through monosynaptic inhibitory inputs to vBNST neurons that project to the lateral hypothalamus. Altogether, the work beautifully describes a novel circuit that mediates inflammation-related suppression of feeding.

The paper is well written, the stats are clear, and the figures are easy to understand. The authors' claims are very well supported, and the appropriate literature is cited. The work is straightforward, but will be certainly be of great interest to researchers interested in feeding behavior. I have no major concerns with manuscript. I commend the authors for completing a great project and only have a few suggestions.

Minor concerns:

1. The involvement of this circuit in loss of appetite is a key strength of the paper. To bolster this relationship, I wish the authors had tested if inhibiting vBNST to LH neurons reduces feeding. The photostimulation experiments done on this pathway are interesting, but not very physiological (due to the synchronicity of photostimulation-induced activity).

Response:

We performed this experiment and found a significant decrease in food intake after inhibiting vBNST to LH pathway (see new data in Supplementary Fig. 11). We injected AAVretro-Cre in LH and Cre-dependent hM4Di in vBNST of wild type mice, then we used CNO to silence the LH-projecting vBNST neurons. The level of feeding suppression is not as strong as activation of ovBNST PKC- δ neurons. We think the possible reasons are: (1) in order to restrict the virus in vBNST, we injected a small amount of virus (~90 nl), so the number of neurons may not be sufficient; (2) unlike optogenetic activation experiments in which activation of a subset of neurons will affect behaviors, the silencing or "loss-of-function" requires loss of majority of the neurons in a given population to take effect; (3) the AAVretro virus may not be efficient enough to label sufficient number of the LH-projecting vBNST neurons, which could be solved when a more efficient AAVretro virus would become available. However, despite all these technical limitations, we still observed a significant decrease of food intake after inhibiting vBNST-LH pathway, which further supported our circuits model.

2. If the authors' tried using excitatory DREADDs to alter feeding behavior, I encourage them to add that data to the manuscript. Since the inhibitory DREADD experiments and excitatory opto

experiments both worked so well, it would be interesting to know if excitatory DREADDs were any less effective in altering feeding behavior. The two approaches cause very different types of elevations in spiking activity.

Response:

We performed the experiments and found that chemogenetic activation of the ovBNST PKC- δ neurons also significantly suppresses food intake (see new data in Supplementary Fig. 4b).

3. It would be nice to see if the vBNST neuron manipulations altered behavior in a general manner (e.g., locomotor activity and anxiety), as was tested with the ovBNST neuron manipulations.

Response:

We have included this data (Supplementary Fig. 10). We did not find significant changes in anxiety levels in the elevated plus maze and open field tests. However, we found that the activation of the vBNST-LH pathway slightly increases the velocity and the total distance moved in the open field test.

4. Supplemental figure 1 should be highlighted in the results section. Panel K is particularly interesting and deserves a mention in the text. Also, do you know what percent of ovBNST neurons are PKC- δ positive?

Response:

We have highlighted the result in the main text. The PKC- δ positive neurons occupy around 50% of the ovBNST neurons based on DAPI and immunostaining with PKC- δ antibody, and also by blind electrophysiological recording of ovBNST neurons (66 out of 114 neurons recorded in ovBNST are PKC- δ positive neurons as identified post-recording).

5. Considering that “act” is already a word, it might not be the best acronym for anterior commissure. I am not sure that an acronym is even necessary here.

Response:

We have changed “act” to “anterior commissure” in the text.

Reviewer #2

In this manuscript, Wang and colleagues have identified a new BNST microcircuit involving PKC-delta expressing neurons, which plays a role in inflammation-associated anorexia and in feeding behavior. The study is original and well performed and the manuscript well written. Statistical analysis is appropriate.

Comments arising from the reading of the manuscript are further detailed below.

Major :

3. Why did the authors think that the PKCdelta subpopulation in ovBNST might be relevant to characterize? The initial hypothesis that guided the authors towards this specific subpopulation is unclear and needs to be addressed in the introduction.

Response:

We thank the reviewer for pointing this out and giving us an opportunity to clarify. Previous studies have suggested that LPS or IL-1 β activates neurons in ovBNST, therefore we searched for genetic markers that could label ovBNST neurons. To dissect the BNST microcircuit, we also searched for several other genetic markers in BNST, including CRF, CCK and somatostatin. We found that PKC- δ is the only marker specifically located in the oval region but not in other nuclei of BNST. We have accordingly revised the text in the introduction section.

1. Is the activation of PKC-delta neurons in ovBNST specific to IL1beta and LPS? What about other important inflammatory signals such as TNFalpha or IL-6? It would be relevant to understand whether these cells respond only to certain inflammatory stimuli, or not.

Response:

We have tested TNF α , which also preferentially activates ovBNST PKC- δ neurons. We also tested cholecystokinin (CCK, IP injection of low dose 5 μ g/kg mimics satiety) and LiCl (induces nausea and visceral malaise). ovBNST PKC- δ neurons are also activated by LiCl but not by CCK selectively (See the data in Supplementary Fig. 2). These data suggest that ovBNST PKC- δ neurons might be specific for pathological or inflammation-associated anorexia. But because there are numerous other factors causing anorexic behavior, it is impossible to test all of them. Therefore, we can conclude that the ovBNST PKC- δ neurons are involved in the anorexia caused by inflammatory stimuli but cannot exclude them from other types of anorexia.

2. Related to the point above: do the authors have any evidence that this sub-group of cells is activated in cancer-bearing or other mice models of inflammation-associated anorexia?

Response:

We agree that it is interesting and important whether the ovBNST PKC- δ neurons are activated in cancer and that they could regulate food intake in cachexia. However, the cancer condition is much more complicated and is beyond the scope of current paper. Campos *et al* (Nat Neurosci 20:934–942) showed that Lewis lung carcinoma (LLC) cell implantation induces c-Fos activation in ovBNST. But whether the activated neurons are selectively in PKC- δ population remains to be determined.

3. The authors report in figure 1D that only 20% of PKC-delta cells are activated in response to inflammatory stimuli. This represents quite a small group of cells after all and this point should be discussed in the discussion section.

Response:

We agree with the reviewer's concern and revised text in the discussion. It also should be noted that this result is based on c-Fos expression, which may not detect all the neurons that are activated.

Minor :

4. What was the success rate of expression of Chr2 and hM4Di in PKC-delta neurons? was this verified? This information should be included in the methods and supplementary figures.

Response:

We usually get more than 90% of the PKC- δ neurons to express Chr2-EYFP or hM4Di-mCherry as verified through post-mortem processing of brain tissue sections in multiple sets of mice, and have included this information in the methods.

2. In the methods, please specify if C57Bl/6 used were Bl6J or Bl6N.

Response:

In order to be consistent with previous studies (Nat Neurosci. 17:1240-8, Nature. 468:270-6), we used the C57BL/6crl, a line from Charles River Laboratory. We have included this information in the methods.

3. In the methods, please give more details on how the double immunofluorescent staining for PKC-delta and c-fos was actually performed. Please also define how many sections per brain and how many brains per condition were evaluated.

Response:

We have revised the method text accordingly.

Reviewer #3

Wang et al., reported that the oval region of the BNST mediated inflammation-associated anorexia by using chemo/optogenetic approaches, and demonstrated a local circuit in the BNST. This study covered both BNST and LH, which have been previously extensively studied in the fields of feeding behavior. The neural mechanisms of inflammation-associated anorexia is relatively unclear, which was interesting. However, there were several major problems which largely reduced my enthusiasm. For example:

1. In the figure 1a, the adjacent areas were too dim to recognize, which are important to clarify the studied brain area (ovBNST);

Response:

We have included a new figure in Supplementary Fig. 1a with increased background in a different color to show the BNST and surrounding areas. One advantage of using genetic markers to label specific type of neurons is to compare c-Fos expression or other structure information across different animals more accurately and more conveniently. In fact, here we can easily use PKC- δ expression to identify ovBNST and check the c-Fos expression in this region.

2. One big concern was the number of inflammation-related Fos-positive neurons in Fig. 1c-d was too small (less than 20). How could this small number of neurons affect behaviors, for example, feeding behavior in this study? This was the "core" part of this study, but it was meaningless.

Response:

We agree with the reviewer's concern and added a short discussion in the text. It should be noted that this result is based on c-Fos expression, which may not detect all the neurons that are activated. It is not totally surprising that manipulation of a small number of neurons can change feeding behaviors. For example, both the ARC AGRP neurons and LPB CGRP neurons are populations of a relatively small number of neurons, yet, both of them have a strong effect on feeding. Betley *et al.* (Cell, 155:1337-50) even showed that activation of a subset of AGRP neurons can strongly induce food intake.

3. Another big concern was why only 20 min food intake was recorded and analyzed? If not affecting long term food intake (ie. hours), it was meaningful either. Moreover, this reviewer was not

comfortable with analyzing feeding behavior by using food intake per body weight. The amount of food intake should be used to perform feeding assays.

Response:

We used the 20 min or 30 min assay for these reasons: (1) make it consistent with a previous study (Nat Neurosci. 17:1240-8); (2) the effect of feeding inhibition and feeding induction are very strong in both the fed and the 24-hr fasted states, 20-30 min is sufficient to detect the significant difference of the food intake change; (3) it is very challenging to couple optic fibers with multiple animals and test feeding in a similar time window of the day, therefore many optogenetic studies test feeding in a short time window of 10-30 min (for example, Nat Neurosci.20:1384-1394, Nat Commun. 9:52). Together, we feel that our testing protocol is appropriate and sufficient to address the questions in this paper. But we agree with the reviewer's concern that food intake over a longer time is also interesting and important. Therefore, we added chemogenetic activation experiments to show that the food intake was also significantly suppressed in 2-hr feeding tests by activating ovBNST PKC- δ neurons (Supplementary Fig. 4b).

We normalized the food intake per body weight because we used both male and female mice in this study. In a pilot test, we found that male and female animals with similar age eat differently mostly due to their different body weight (Supplementary Fig. 3). When we normalized the food intake to their body weight, there was no difference between male and female animals. We have added the food intake data to clarify this (Supplementary Fig. 3). Furthermore, many previous studies measured food intake per body weight to study feeding behavior (for example, J Neurosci. 23:10084-92, J Clin Invest. 103: 383–391; Nat Neurosci 17:667–669).

4. Following #3, 24 hr food deprivation would stress animals, so physiological feeding assays (ie. feeding in later afternoon or evening or night feeding should be repeated by using similar manipulations.

Response:

We totally agree with the reviewer's concern. That's why we performed the feeding tests at both 24-hr fasted and fed states in almost all the experiments. The tests at fed state were performed in the later afternoon. To minimize the stress effect and further confirm the feeding suppression effect after activation of ovBNST PKC- δ neurons, we also tested the feeding in their home cages (Fig. 2h-j).

5. In Fig. 2, feeding assays should be performed on same mice before, during and after photostimulation. Otherwise, individual variations in feeding would bring "off-target" effects.

Response:

We agree that individual variations could bring some effects. However, this effect can be cancelled or minimized by recruiting sufficient number of animals, which is what we did in this study. While performing feeding on the same mice before, during and after photostimulation is good for induced feeding by optogenetics, it is challenging to test feeding suppression effect because mice usually show a decrease in feeding over the time. This protocol is not possible for chemogenetic methods. Furthermore, one concern of this protocol is light itself might affect feeding behavior, which will affect feeding suppression effect. One of our major findings is that optogenetic activation of the ovBNST PKC- δ neurons suppresses food intake while chemogenetic silencing these neurons increases food intake. Thus, we cannot use the suggested protocol for food intake measurement in this study. Our feeding interruption experiments in the home cage (Fig. 2h-j) actually used a protocol similar to that the reviewer suggested, and we included EYFP controls to rule out the light effect.

Reviewer #4

The manuscript by Wang et al. describes a new BNST microcircuit that suppresses food uptake in response to satiety and inflammatory signals. Using c-fos immunoreactivity as a marker for neuronal activity, the authors identify in the oval region of the BNST a subpopulation of protein kinase C-delta expressing neurons, which become activated by inflammatory signals such as IL-3beta or LPS. Chemogenetic silencing of ovBNST-PKCdelta neurons attenuated the anorexia induced by IL-3beta or LPS, and increased food intake of fed and fasted mice. Optogenetic activation of ovBNST-PKCdelta neurons produced the opposite effect. ovBNST-PKCdelta neurons send inhibitory projections to neurons in a ventral-lateral subregion of the BNST, and these vlBNST neurons project to the lateral hypothalamus to promote feeding. These LH-projectors are downstream of the ovBNST-PKCdelta neurons, because they can overcome the feeding block induced by ovBNST-PKCdelta neuron activation. Finally, the authors use monosynaptic Rabies tracing to map the inputs of ovBNST-PKCdelta neurons.

I find this study interesting and the data largely convincing. The authors used state-of-the-art circuit mapping tools to discover a novel role of BNST neurons in regulating feeding. I only have minor points, which the authors could easily address without performing additional experiments.

1. It is unusual that food consumption is normalized to the body weight of the animal. Considering that all the results were normalized in that way, I am wondering whether the authors could show that food consumption actually correlates with BW in their experimental groups.

Response:

Please see the response to Review #3 point 3, 2nd paragraph.

2. In figure 2i, it is interesting to note that although the feeding bouts are shorter because they are interrupted by light activation, they are more numerous. This suggests that while activation of ovBNST-PKCdelta cells suppresses food intake, it does not appear to impair the hunger feeling nor the motivation to consume food. The authors may want to comment on this in the manuscript.

Response:

We thank the reviewer's thought on this result, and have added comments on this result accordingly.

3. In figure 5b,c, the legend does not provide information on the duration of the experiment.

Response:

We thank the reviewer for pointing this out, we have added the information in the methods to avoid redundancy.

4. In figure 6g, the fraction of AGRP+ dsRed+ cells was not quantified. The authors should quantify or state why it was impossible to do so.

Response:

We included a comment in the results section to explain that robust quantification is not possible because the AGRP antibody cannot stain cell body robustly.

We thank all the reviewers for their perceptive and constructive comments, and hope that with these revisions and additional data, the paper will be acceptable for publication in *Nature Communications*.

Reviewers' comments:

Reviewer #1 (Remarks to the Author):

The authors nicely addressed all of my comments and concerns. I am happy with the revised manuscript and support publication. This work nicely demonstrates that PKC- δ neurons in the ovBNST regulate inflammation-induced anorexia and exert bidirectional control over feeding behavior more generally. Altogether, the work beautifully describes a novel circuit that mediates inflammation-related suppression of feeding.

Reviewer #2 (Remarks to the Author):

The authors have satisfactorily addressed the comments of this reviewer.

Reviewer #3 (Remarks to the Author):

The authors improved the manuscript, but did not substantially address this reviewer's original concerns, which were critical to draw the conclusions of this study. Additional experiments should be performed to address them.

For example, for my original concern#2, the authors responded with " It should be noted that this result is based on c-Fos expression, which may not detect all the neurons that are activated". If so, it would be interesting and necessary to know how many neurons were indeed activated in the experimental conditions and how to accurately count the activated neurons.

Also, for my concern#3, the authors mentioned both male and female mice were used in the study, which was the reason for the authors to use food intake at gram per body weight. This reviewer did not agree on this argument. Even the data from both male and female mice were grouped together, the absolute food intake on males and females should also be significant with the manipulations if it was true. More importantly, the data from males and females should first be analyzed separately and compared to each other before pooling together. If there was difference between males and females, the data should be plotted and shown separately.

Also, I was not comfortable with the responses to my original concern#4 and 5, which should also be substantially addressed, to this reviewer, before it could move forwards.

Reviewer #4 (Remarks to the Author):

The authors addressed all my (minor) issues in the revised manuscript. I have no further requests for changes and consider the manuscript ready for publication as it is.

Response to reviewers: Wang et al.

We would like to thank the reviewers for their excellent and insightful comments. We have added new data and text revision in response to reviewers' comments. Please find below our response detailing these changes.

Reviewer #1 (Remarks to the Author):

The authors nicely addressed all of my comments and concerns. I am happy with the revised manuscript and support publication. This work nicely demonstrates that PKC- δ neurons in the ovBNST regulate inflammation-induced anorexia and exert bidirectional control over feeding behavior more generally. Altogether, the work beautifully describes a novel circuit that mediates inflammation-related suppression of feeding.

Reviewer #2 (Remarks to the Author):

The authors have satisfactorily addressed the comments of this reviewer.

Reviewer #3 (Remarks to the Author):

The authors improved the manuscript, but did not substantially address this reviewer's original concerns, which were critical to draw the conclusions of this study. Additional experiments should be performed to address them.

For example, for my original concern#2, the authors responded with " It should be noted that this result is based on c-Fos expression, which may not detect all the neurons that are activated". If so, it would be interesting and necessary to know how many neurons were indeed activated in the experimental conditions and how to accurately count the activated neurons.

Response:

Although there is some limitation as we suggested in our previous response, the expression of immediate-early gene such as c-Fos is the current most accurate method for screening all the neurons in a specific brain region after a stimulus. Therefore, we used this method to demonstrate that ovBNST PKC- δ neurons are activated by the inflammatory signals. The point we want to conclude is that the inflammatory signals preferentially activate ovBNST PKC- δ neurons but not ovBNST PKC- δ negative neurons, for which the method we used is sufficient and appropriate. More importantly, we performed chemogenetic silencing experiment to demonstrate that silencing these neurons effectively attenuates the feeding suppression caused by those inflammatory signals, which is a more direct way to demonstrate the role of these neurons in this condition.

Also, for my concern#3, the authors mentioned both male and female mice were used in the study, which was the reason for the authors to use food intake at gram per body weight. This reviewer did not agree on this argument. Even the data from both male and female mice were grouped together, the absolute

food intake on males and females should also be significant with the manipulations if it was true. More importantly, the data from males and females should first be analyzed separately and compared to each other before pooling together. If there was difference between males and females, the data should be plotted and shown separately.

Response:

We thank the helpful suggestions by the reviewer. We have showed the absolute values of food intake in both male and female animals in optogenetic activation experiments (Supplementary Fig. 4c), chemogenetic silencing experiments (Supplementary Fig. 6a) and animals injected with saline or CNO (Supplementary Fig. 3a). The data were analyzed separately for male and female mice. These results demonstrated that food intake was suppressed in both male and female mice after optogenetic activation of ovBNST PKC- δ neurons (Supplementary Fig. 4c) and increased in both male and female mice after chemogenetic silencing (Supplementary Fig. 6a). As suggested by the reviewer, we also analyzed the absolute food intake data by pooling male and female animals together (Supplementary Fig. 4d, and Supplementary Fig. 6b), which also confirmed our conclusions.

Also, I was not comfortable with the responses to my original concern#4 and 5, which should also be substantially addressed, to this reviewer, before it could move forwards.

Response:

For points #4, the food intake tests at fed state were performed in the later afternoon as suggested by the reviewer. To minimize the stress effect and further confirm the feeding suppression effect after activation of ovBNST PKC- δ neurons, we also tested the feeding in their home cages (Fig. 2h-j).

For points #5, we have added the baseline measurements of individual food intake data and compared that after light stimulation in the same individual animals (Supplementary Fig. 4a, b). These within-subject comparison data again strongly supported our conclusion that optogenetic activation of ovBNST PKC- δ neurons suppresses food intake.

Reviewer #4 (Remarks to the Author):

The authors addressed all my (minor) issues in the revised manuscript. I have no further requests for changes and consider the manuscript ready for publication as it is.

REVIEWERS' COMMENTS:

Reviewer #3 (Remarks to the Author):

The authors addressed my concerns and I do not have additional questions.